# Production of a Phosphate Concentrate from the Tailings of a Niobium Ore Concentrator

**Anthony Clapperton [1], Claude Bazin [2,*], Dominic Downey [3] and Jean-Sébastien Marois [3]**

1    Soutex, Québec, QC G1N 4C4, Canada
2    Department of Mining, Metallurgical and Materials Engineering, Laval University,
     Québec, QC G1V 0A6, Canada
3    Niobec Mine, St-Honoré, QC G0M 1V0, Canada; Dominic-D@niobec.com (D.D.);
     JeanSebastien-M@niobec.com (J.-S.M.)
*    Correspondence: claude.bazin@gmn.ulaval.ca

**Abstract:** Apatite is the main source of phosphorous for the making of chemical fertilizers. While apatite is usually recovered from phosphate orebodies as the primary product of a mining exploitation, this paper documents the approach taken to produce a phosphate concentrate as a secondary product from the tailings of a niobium ore concentrator. The conventional desliming/flotation scheme used to process phosphate ores was tested and adapted to process one of the reject streams of a niobium concentrator in order to produce a salable phosphate concentrate. For that particular material, it was found that the reverse flotation of apatite yielded better results than the commonly used direct flotation of apatite. The recommended approach to produce the phosphate concentrate is desliming followed by reverse flotation of apatite and an acid leaching of the apatite concentrate to lower the MgO content below the specification for a phosphate concentrate. The obtained phosphate concentrate assays more than 32% $P_2O_5$ at a $P_2O_5$ recovery of 41%, which although low is found to be economic for the case of processing plant reject tailings.

**Keywords:** apatite; direct flotation; reverse flotation; desliming; leaching

## 1. Introduction

Apatite ($Ca_5(PO_4)_3(F,Cl,OH)$) is the main source of phosphorous for the making of chemical fertilizers as discussed in [1]. Apatite is usually recovered from sedimentary orebodies and/or from igneous orebodies [2–5]. The ore processing scheme for igneous orebodies is conventional and consists of grinding followed by flotation [2,6]. The beneficiation of apatite from igneous orebodies implies the mining and size reduction of the ore, which represent a significant, if not the main, contribution to the operating costs of the process. Apatite is often associated to ores exploited for other economical minerals, as it is the case for iron ores. For such ores, apatite is considered as a non-valuable mineral and it is discarded as gangue to the tailings ponds of the concentrator. The reprocessing of tailings with the objective of recovering other species than the species targeted by the mineral processing plant is not a subject discussed abundantly in the literature [7], although it is recognized [8] as an approach to reduce the environmental footprint of a mining exploitation by reducing the mass of material discarded into the tailings ponds [7]. Reprocessing of tailings also allows to remove from the tailings species that could be detrimental for the environment, as it is the case of phosphate minerals that can release harmful phosphates [8] for the aquatic life. This paper assesses a process to recover apatite from the tailings of a niobium concentrator into a salable phosphate concentrate.

Many papers deal with the separation of apatite by flotation from carbonate ores [2–6,9]. All these papers examine the apatite concentration process from a freshly ground ore to obtain a phosphate

concentrate as a primary product. Only one paper was found to deal with the production of a phosphate concentrate from the tailings of a Brazilian fertilizer plant [7], but it was not possible to find any paper describing the recovery of apatite from the tailings of a mineral processing plant which is the subject of this paper. The processing scheme usually retained for phosphate ore is grinding, desliming [2,4] to remove fine particles, followed by a direct [3] or reverse [2] flotation of apatite. A good review of the flotation practices and reagents used for phosphates ores is given in [10]. This paper shows that it is possible to apply the conventional apatite ore processing scheme to produce a salable phosphate concentrate from a tailing stream of a niobium ore concentrator that recovers pyrochlore and rejects apatite with the other gangue minerals to tailings.

The successful validation and commissioning of a process to produce a phosphate concentrate from the tailings of the concentrator would reduce the plant footprint by reducing the mass of the concentrator reject by approximately 7%, in addition to generate secondary revenues from the selling of the concentrate. This paper also demonstrates the feasibility of recovering apatite minerals that have been in contact with flotation reagents used to recover other minerals.

## 2. Materials and Methods

### 2.1. Source of the Tailings for the Beneficiation of Apatite

The studied mine tailings come from the Niobec mine located approximately 20 km north of Ville Saguenay, QC, Canada. The mined ore comes from a carbonatite formation containing pyrochlore $(Na, Ca)_2Nb_2O_6(OH, F)$ and columbite $(FeO \cdot Nb_2O_5)$ as the main niobium carriers. The major gangue minerals (~65%) are calcite $(CaCO_3)$, dolomite $((Ca,Mg)CO_3)$, and ankerite $(Ca(Fe,Mg,Mn)(CO_3)_2$, followed by apatite (at 35%) and silicates (zircon, biotite, chlorite, and feldspath). Figure 1 shows the ore processing circuit [11,12].

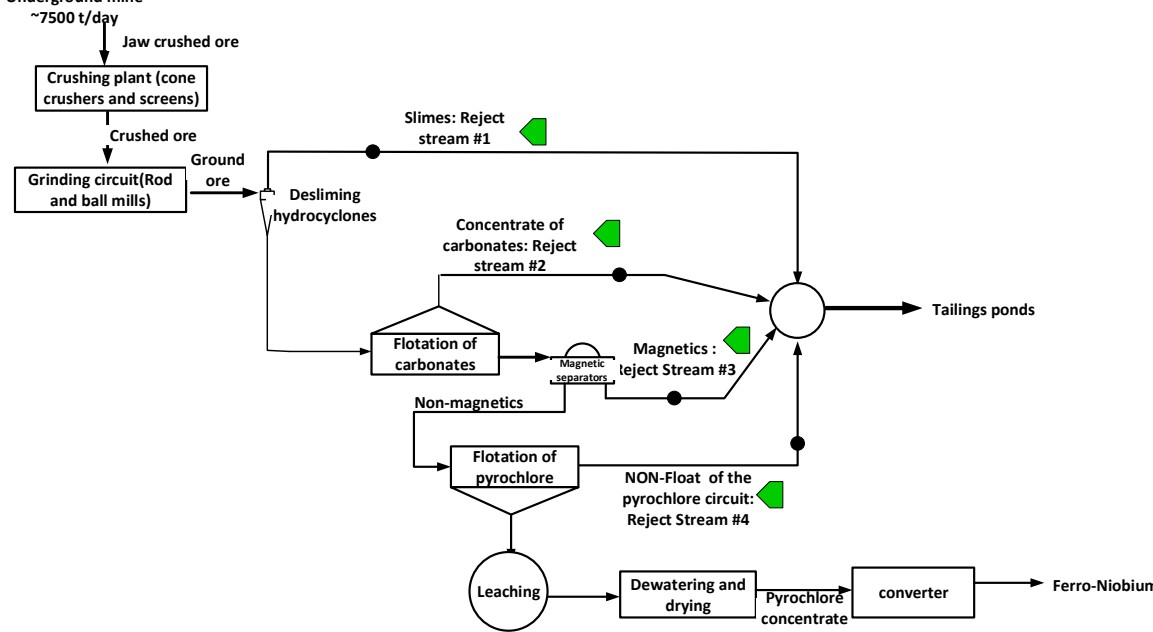

**Figure 1.** Simplified flowsheet of the Niobec concentrator showing the four reject streams.

The mined ore is first jaw and cone crushed. The crushed ore feeds the grinding circuit that consists of a rod mill followed by a ball mill. The ground product feeds desliming hydrocyclones that remove fine particles or slimes. The slimes (−10 μm) are directed to the tailings ponds as a first reject stream. The coarse fraction of the desliming hydrocyclones feeds a carbonate flotation circuit whose role is to float most of the carbonate minerals of the ore. The floated Carbonate Concentrate

(CC) is directed to the tailings ponds as the second reject stream of the concentrator. The non-floated material from the carbonate circuit passes through magnetic separators. The magnetic material (mainly magnetite ($Fe_3O_4$)) is sent to the tailings ponds as the third reject stream, while the non-magnetic material advances to the pyrochlore flotation circuit. The floated pyrochlore concentrate is upgraded by leaching soluble species, dried, and sent to the converter that produces the ferro-niobium for the steel industry. The non-floated material from the pyrochlore flotation circuit joins the other three reject streams as the fourth reject stream. The circuit also incorporates flotation banks not shown in Figure 1 to remove sulfide minerals from the pyrochlore concentrate. The detailed flowsheet is available in [11].

The characteristics of the four reject streams are summarized in Table 1. The data of Table 1 is calculated from the daily sampling and production balances of the concentrator. Most (45.5%) of the apatite is carried by the reject of the pyrochlore flotation circuit. The highest apatite concentration (5.1% $P_2O_5$) is observed for the CC or reject stream #2 in Figure 1. The 5.1% $P_2O_5$ is comparable to the ore head grade of some apatite igneous projected and/or operated mines [1–5,9]. In the case of the considered material, the ore is already mined and ground which provides a significant economic advantage over conventional processes for the production of a phosphate concentrate from ores. Because of its phosphate content, the CC stream is selected as the most promising reject stream to recover apatite. Two options are then available:

1.  To produce a phosphate concentrate to be spread directly on top of the agricultural fields [13].
2.  To maximize the plus-value of the product by producing a phosphate concentrate for the market of chemical fertilizers.

**Table 1.** Characteristics of the reject streams from the Niobec concentrator [11,12].

| Stream | Reject Stream # | Weight Distribution from Mill Feed (%) | Apatite Distribution from Mill Feed (%) | $P_2O_5$ Content * (%) |
|---|---|---|---|---|
| Slimes | 1 | 13 | 10 | 2.3 |
| CC | 2 | 25 | 42.5 | 5.1 |
| Magnetic concentrate | 3 | 3 | 0.4 | 0.43 |
| Reject of the pyrochlore circuit | 4 | 58 | 45.5 | 2.3 |

* The phosphate ($P_2O_5$) content is measured by X-ray fluorescence (XRF) and it is used as an indirect measurement of the apatite content.

If there is practically no constraint on the composition for the phosphate concentrate of the first option, the production of a commercial concentrate implies to verify the composition constraints identified in Table 2.

**Table 2.** Composition constraints for a commercial phosphate concentrate [14,15].

| Species | Acceptable Range |
|---|---|
| $P_2O_5$ | >30% |
| %CaO/%$P_2O_5$ | <1.6 |
| MgO | <1.0% |
| $Al_2O_3$ | <5.0% |
| $Fe_2O_3$ | ~2–3% |
| $SiO_2$ | ~2.0% |
| Cl | <0.1% |
| Na-K | 0.1–0.8% |

*2.2. Sample Preparation and Characterization, Methods of Analysis*

A 40 kg sample of the CC (see Figure 1) was prepared by the Niobec personnel by combining the daily production samples collected during one week in order to have a sample as representative as possible of the typical material flowing in the CC stream. Indeed, during a typical day of operation,

the plant operators take, on a 4 h basis, a cut containing ~900 g of solids from each strategic stream of the circuit including the CC. For the project, the operators were asked to collect two cuts from the CC, one for the regular daily production balance of the mill and the other one for the project on the recovery of apatite. The cuts taken for the apatite project were composite in the same recipient during one week. Therefore, the sample collected during one week consists of $6 \times 7 = 42$ cuts taken at different hours of different days and is considered sufficiently representative of the CC to carry out the experimental testwork aiming at producing an apatite concentrate from this material. In the preparation for the testwork to be conducted, the composite sample was kept wet, homogenized with a mixer, and was shipped wet to the laboratory in order to minimize any degradation of the collector already adsorbed on the surface of the carbonates.

### 2.2.1. Sample Characterization

The wet CC sample was filtered to remove most of the water and the wet cake was split on the filter cloth by taking random quarters of the cake. A 2 kg sample was dried and prepared for chemical, X-ray diffraction (XRD), and size analyses. The reject sample is kept wet for the flotation tests as shown in Figure 2.

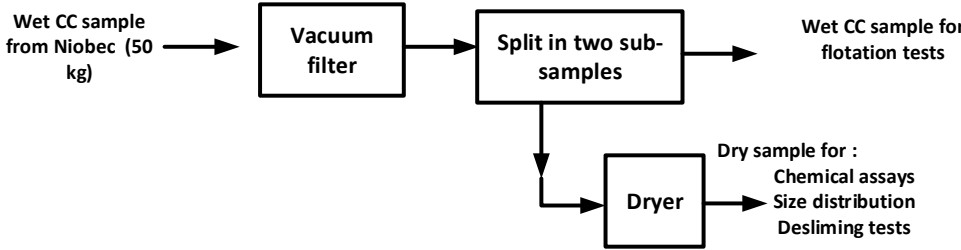

**Figure 2.** Sample preparation.

### 2.2.2. Analysis Instruments

The solid samples are assayed using an Epsilon 1 Energy Dispersive X-Ray Fluorescence (ED-XRF, Malvern Pananalytical, Toronto, ON, Canada). XRF samples are fused with lithium Borax to prepare glasses for the chemical analysis. The relative assaying errors are 4% for $P_2O_5$, 5% for MgO, and CaO, and 8% for $SiO_2$ and elements (e.g., Fe, Al, S, Na, and K) whose contents are below 3%. These errors are used as weighing factors for the mass balancing or data reconciliation [16] of the experimental results. Sieving is done in two steps. The sample is firstly washed on a 400 mesh screen. The +400 mesh is dry sieved using Tyler screens ranging from 150 to 38 μm (100 to 400 mesh in a $\sqrt{2}$ geometric progression).

### 2.2.3. Flotation Equipment

All the rougher flotation tests were conducted using a laboratory 2.5 L Denver Sub-A flotation cell. A 1.0 L cell for the cleaning of the rougher concentrate produced in the case of the direct flotation tests. The flotation reagents and operating conditions are identified in the Section 3.2 as these differ for the direct and inverse flotation tests.

### *2.3. Mineral Composition and Phosphate Content as a Function of Particle Size*

Table 3 gives the mineral composition estimated from elemental assays using the method proposed by Whiten [17]. As anticipated, the CC consists largely of carbonates, mainly dolomite and calcite at 40% with less than 10% ankerite. The «*other*» minerals in Table 3 are iron oxides, zircon, and sulfides (<2%). The apatite found in the tailings is a magmatic type fluor-apatite with the following composition of 36% Ca, 42% O, 2.2% F, 0.3% Na, 1.1% Mg, 0.4% Si, 15.5% P, 0.2% Mn and 1.6% Fe, and 1.0% Sr as measured by SEM [18]. There were no radioactive elements detected in the apatite.

**Table 3.** Estimated mineral contents of the CC (average ± standard deviation of 10 measurements).

| Mineral | % |
|---|---|
| Calcite | 36 ± 5 |
| Dolomite | 39 ± 8 |
| Ankerite | 9 ± 5 |
| Total carbonates | 84 ± 15 |
| Apatite | 12 ± 3 |
| Silicates | 2 ± 0.7 |
| Other | 2 ± 1.5 |

The size distribution of the CC with the phosphate content of the material retained within each size fraction is given in Table 4. Approximately 25% of the phosphate minerals are contained in the +38 μm, which provides an incentive to remove the −38 μm from the CC, prior to the concentration process.

**Table 4.** Phosphate contents within the size intervals of the concentrate of carbonates (average ± standard deviation of 5 measurements from 5 random samples of the CC).

| Particle Size (μm) | % Retained | % $P_2O_5$ | Cumulative % $P_2O_5$ | Distribution % |
|---|---|---|---|---|
| Head | 100 | 5.0 ± 1.2 | | |
| 106/150 | 1 ± 0.4 | 25.5 ± 3.2 | 25.5 ± 3.2 | 5 ± 1 |
| 75/106 | 3.3 ± 1.1 | 17.7 ± 1.6 | 19.9 ± 2.9 | 11 ± 2 |
| 54/75 | 9.5 ± 1.2 | 7.9 ± 1.1 | 11.4 ± 2.1 | 15 ± 4 |
| 38/54 | 29 ± 3.1 | 4.3 ± 0.9 | 6.6 ± 1.8 | 25 ± 7 |
| <38 | 57.4 ± 5.2 | 3.8 ± 0.8 | | |

## 3. Experimentation and Results

As indicated above, the objective of the test work is not to design a new process to concentrate apatite, but to assess if the conventional desliming/flotation processing scheme for phosphate orebodies can be successfully adapted to produce a phosphate concentrate from the tailings stream of a concentrator. Furthermore, all the following testing was done with the objective of producing a salable concentrate. It is not a study with an objective of optimizing an existing process. Once a process would have been demonstrated as capable of producing a salable phosphate concentrate, the testing will continue with the objective of maximizing the recovery of apatite and minimizing the operating costs associated to the process.

### 3.1. Pre-Concentration of Apatite

The data of Table 4 shows that a split of the carbonate concentrate (CC) at a 38 μm size could increase the phosphate ($P_2O_5$) content from 5.1% (total) to 6.7% (+38 μm) at the cost of losing 75% of the phosphate. Such losses may be unacceptable for a fresh ore, but as the CC is already sent to tailings, it can be afforded here. The removal of fines in apatite ore processing is a standard procedure [1–5,9] as fines consume reagents and cause a loss of selectivity during the subsequent flotation step used to concentrate apatite. The desliming of the CC has been assessed with the dry sub-sample (see Figure 2) using a hydrocyclone and a spiral concentrator. Table 5 summarizes the results obtained with the hydrocyclone and the spiral. As suggested by the size distribution data of Table 4, the optimum cut size should be ~38 μm, which would require a 10 cm diameter hydrocyclone that was not available at the time of the test work. Preliminary tests were then conducted using a 5 cm diameter hydrocyclone [18]. The observed results (Table 5) show a marginal increase in the phosphate content from the feed (5.1% $P_2O_5$) to the underflow (5.9% $P_2O_5$). Apatite recovery to the hydrocyclone underflow is only 16%, but this value could certainly be increased via some optimization work to find an adequate hydrocyclone diameter and configuration (apex, vortex finder, inlet pressure … ). A new set-up is currently being installed to carry out the tests with hydrocyclones of various geometries. At this early stage of the process development, the main objective was to find the method that can yield the highest

upgrading of $P_2O_5$, keeping in mind that if a downstream apatite upgrading process can be designed to achieve the production of a phosphate concentrate satisfying the constraints of Table 2, the tuning of the desliming method can be revisited with the objective of maximizing the recovery of apatite.

**Table 5.** Results of the pre-concentration using the hydrocyclone and the spiral concentrator ($P_2O_5$ and MgO contents and distributions are average values of 4 repetitions ± the standard deviation).

| Flux | Hydrocyclone Desliming * (Statistics on 4 Replicate) | | | | | Spiral Desliming ** (Statistics on 4 Replicates) | | | | |
|---|---|---|---|---|---|---|---|---|---|---|
| | Weight (%) | %$P_2O_5$ | $P_2O_5$ Distribution (%) | %MgO | MgO Distribution (%) | Weight (%) | %$P_2O_5$ | $P_2O_5$ Distribution (%) | %MgO | MgO Distribution (%) |
| Feed | 100 | 5.1 ± 0.8 | 100.0 | 12.3 ± 1.2 | 100.0 | 100.0 | 5.62 ± 0.5 | 100.0 | 12.5 | 100.0 |
| Reject | 85.8 | 5.0 ± 1.0 | 83.6 ± 0.4 | 12.5 ± 1.3 | 87.2 ± 0.9 | 84.7 | 5.4 ± 0.5 | 81.7 ± 2 | 12.7 ± 0.8 | 85.9 ± 1.1 |
| Pre concentrate | 14.2 | 5.9 ± 0.8 | 16.4 ± 3.0 | 11.1 ± 0.9 | 12.82 ± 0.7 | 15.3 | 6.7 ± 0.7 | 18.3 ± 0.7 | 11.5 ± 0.9 | 14.1 ± 0.5 |

*: The pre-concentrate is the hydrocyclone underflow; **: The spiral pre-concentrate is the combined heavy and middlings streams.

The spiral concentrator used to assess the CC pre-concentration is a 5 turn WWW-Plus spiral from Mineral Technologies(Perth, Western Australia) with a diameter of 700 mm [19–21]. The set-up for the test is similar to the one used by Liu et al. [21] and Sadeghi et al. [19]. The slurry feed rate to the spiral is 30 L/min at a solids concentration of 13% w/w. Twenty (20) L/min of wash water is added to the spiral, and it is only after the third turn that the inner flowing slurry is collected by the cutters (Figure 3) to give the deslimed material. The operating conditions were found during a preliminary test by a visual observation of the color of the radial stream bands flowing down the spiral (Figure 3) as the apatite minerals exhibit a darker color than the carbonates. The selected operating conditions are those that yielded the largest inner flowing dark band. Once the operating conditions were established, the spiral was operated until steady state is reached and 4 sampling runs were conducted to obtain samples of the spiral concentrate (apatite pre concentrate) and of the spiral reject. The samples were prepared and assayed separately to obtain the results of Table 5. The 1.31 upgrading ratio (the upgrading ratio is the ratio of $P_2O_5$ content of the pre-concentrate to the $P_2O_5$ content of the feed) obtained with the spiral is comparable to the 1.2 ratio achieved with the hydrocyclone. The idea of testing the spiral is inspired from the work of Liu et al. [22] and was motivated by the fact that apatite is slightly denser and coarser (see Table 4) than the carbonates and it was felt that the spiral could yield a better cut than the hydrocyclone. However, the spiral used for the evaluation is designed for processing coarse material (>1000 μm), while the CC is finer than 150 μm. The desliming performances could certainly be improved by using a spiral designed to process fine particles such as a VHGS from Mineral Technologies.

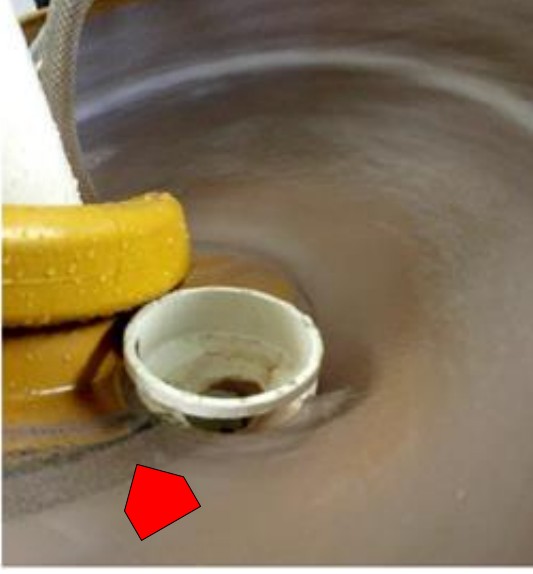

**Figure 3.** Apatite captured by a cutter of the spiral.

The only approaches considered for the desliming of the CC are the hydrocyclone and the spiral, as the two equipments have low operating costs, are not expensive, have no mechanical or moving parts, are robust, and provide a good capacity per unit of floor used. A Knelson concentrator enhanced gravity separator was also tested but did not yield significantly better results than the spiral or the hydrocyclone and that option was not retained for the processing of the CC because the operation of the Knelson concentrator is far more demanding than that of the spiral or hydrocyclone.

### 3.2. Apatite Concentration by Flotation

As more than half of the phosphate is produced by flotation [3,23], only flotation was considered here as a method to concentrate the apatite from the CC. Two processing approaches are industrially used [2]:

- Direct flotation of apatite after the depression of the gangue minerals [2,3].
- Reverse flotation for which apatite is depressed and the gangue minerals are floated [23].

The two approaches are assessed with the deslimed CC samples obtained by washing the wet sample (see Figure 2) on a 400 mesh (0.038 mm) screen [18]. However, as the CC contains minerals that were already floated (see Niobec flowsheet of Figure 1), i.e., for which the surface is hydrophobic, it was anticipated that the reverse flotation approach has a better chance of success than direct flotation. The success of reverse flotation relies on the possibility to find an adequate recipe to selectively depress apatite and take advantage of the fact that the accompanying carbonates are already hydrophobic. Figure 4 shows the test procedure and conditions for direct and reverse flotation. The flotation testing is done using a 2.5 L Denver Sub-A flotation cell for roughing and a 1.0 L cell for the cleaning of the rougher concentrate produced by the direct flotation tests. As anticipated, the reverse flotation yielded better results [18] as summarized in Tables 6 and 7.

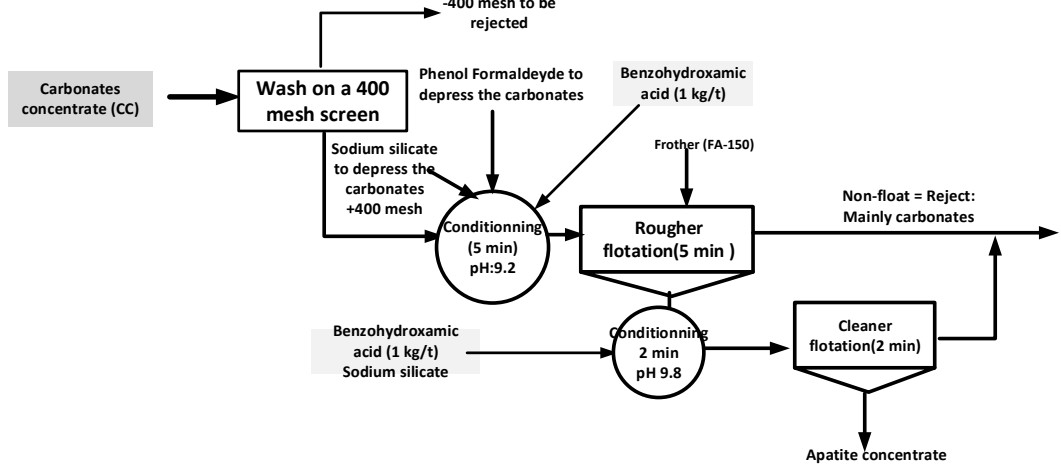

(**a**) Direct flotation.

**Figure 4.** *Cont.*

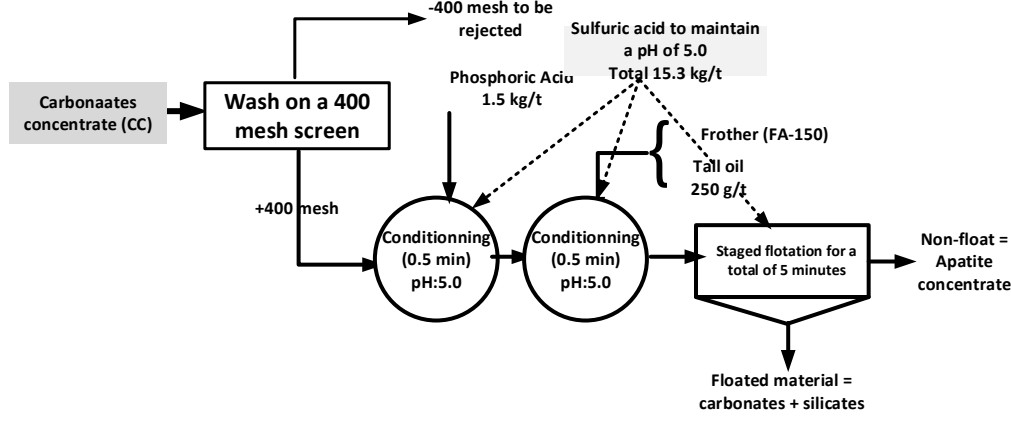

(**b**) Reverse flotation.

**Figure 4.** Test procedures used for direct and reverse flotation of apatite. (**a**) Direct flotation; (**b**) Reverse flotation.

**Table 6.** Results of the direct flotation tests with one cleaning stage.

| Stream | mass (g) | Content (%$w/w$) | | | | | | Distribution (%) | | | | | |
|---|---|---|---|---|---|---|---|---|---|---|---|---|---|
| | | MgO | Al$_2$O$_3$ | SiO$_2$ | P$_2$O$_5$ | CaO | Fe$_2$O$_3$ | wt.% | MgO | Al$_2$O$_3$ | SiO$_2$ | P$_2$O$_5$ | CaO | Fe$_2$O$_3$ |
| Feed | 450.0 | 10.75 | 0.04 | 0.07 | 7.74 | 35.6 | 4.4 | 100 | 100 | 100 | 100 | 100 | 100 | 100 |
| Rougher concentrate | 173.3 | 8.73 | 0.06 | 0.01 | 17.13 | 35.3 | 3.07 | 38.5 | 31.3 | 57.8 | 5.5 | 85.2 | 38.2 | 26.9 |
| Cleaner concentrate | 49.5 | 8.18 | 0.07 | 0.01 | 21.43 | 35.5 | 2.75 | 11.0 | 8.4 | 19.3 | 1.6 | 30.5 | 11.0 | 6.9 |
| Rougher taillings | 276.8 | 12.01 | 0.02 | 0.1 | 1.86 | 35.8 | 5.23 | 61.5 | 68.7 | 30.8 | 87.9 | 14.8 | 61.8 | 73.1 |
| Cleaner taillings | 123.8 | 8.95 | 0.06 | 0.01 | 15.42 | 35.2 | 3.2 | 27.5 | 22.9 | 41.3 | 3.9 | 54.8 | 27.2 | 20.0 |

**Table 7.** Results of reverse flotation of apatite from the deslimed CC sample.

| Stream | mass (g) | Content (%) | | | | | | Distribution (%) | | | | | |
|---|---|---|---|---|---|---|---|---|---|---|---|---|---|
| | | MgO | Al$_2$O$_3$ | SiO$_2$ | P$_2$O$_5$ | CaO | Fe$_2$O$_3$ | wt.% | MgO | Al$_2$O$_3$ | SiO$_2$ | P$_2$O$_5$ | CaO | Fe$_2$O$_3$ |
| Feed | 525 | 11.46 | 0.05 | 0.15 | 8.62 | 33.38 | 4.45 | 100 | 100.0 | 100.0 | 100.0 | 100.0 | 100.0 | 100.0 |
| Rougher floated material | 370 | 14.86 | 0.01 | 0.09 | 0.75 | 33.95 | 5.46 | 70.5 | 91.4 | 14.1 | 42.3 | 6.1 | 71.7 | 86.5 |
| Rougher non-floated material | 155 | 3.33 | 0.15 | 0.29 | 27.43 | 32.02 | 2.04 | 29.5 | 8.6 | 85.9 | 57.7 | 93.9 | 28.3 | 13.5 |

### 3.2.1. Direct Flotation of Apatite

The direct flotation tests use phenol formaldehyde resin (Plenco 14845 Resol brand) and sodium silicate to depress the carbonates as suggested by Bulatovic [24]. Hydroxamic acid (trade name AERO 6493 from Cytec) and benzo hydroxamic acid (trade name Florrea 7510 from Shenyang Florrea Chemicals) [18] were used as apatite collectors for the direct flotation tests. These collectors were found more selective for apatite than the tall oil or fatty acid collectors commonly used [2–6,9]. The conditioning pH is adjusted to 9.8 with NaOH. A rougher concentrate is pulled for 5 min, conditioned with hydroxamic acid and sodium silicate, and cleaned for 2 min of flotation (see Figure 4). Table 6 gives the best direct flotation test results obtained from a series of seven (7) tests [18]. The cleaner concentrate assays 21.43% P$_2$O$_5$, which is low compared to the 30%P$_2$O$_5$ target (Table 2) for a commercial concentrate. The phosphate recovery from the rougher feed is 30.5%. These results are significantly lower than those reported elsewhere [9] for direct flotation. This is due to the uncontrolled flotation of carbonate minerals whose surfaces are already hydrophobic in the CC. The main contaminant of the floated apatite is dolomite that is responsible for the 8.18% MgO content of the cleaner concentrate.

### 3.2.2. Reverse Flotation of Apatite

For the reverse flotation of apatite, sulfuric acid is added to maintain a pH of 5.0 during the conditioning and flotation stages. Phosphoric acid is used to depress the apatite. The consumption of sulfuric acid is high at 15.3 kg/t due to the carbonates that neutralize the acid. Tall oil (Commercial

name Sylfat FA-2) is used as a collector for the carbonates [2–6,9]. Table 7 gives the results of the best performance achieved out of eight inverse flotation tests. The non-floated material from the rougher is the phosphate concentrate which assays 27.4% $P_2O_5$. This phosphate content of the apatite concentrate is achieved with only one flotation stage compared to the 21.4% $P_2O_5$ concentrate obtained from two direct flotation stages (see Table 6). The phosphate recovery is 93% for the reverse flotation compared to 30.5% for direct flotation. From these results, the use of reverse flotation is recommended for the particular application of apatite beneficiation from the floated carbonates stream of the Niobec concentrator. In addition to the better metallurgical results, the advantages of the reverse flotation option over direct flotation for the treatment of the CC are listed below.

- A lower demand and cost for the flotation reagents, particularly the collector used (i.e., benzohydroxamic acid) for the direct flotation of apatite.
- A better rejection of MgO from the apatite concentrate.
- A simple flowsheet (lower Capex) with only one flotation stage.

If the industrial partner decides to carry on with the test work, a Design of Experiment (DOE) as in [6] will be defined and used to optimize the flotation conditions. Indeed, as the aim of the experimentation described here was to demonstrate the feasibility of applying a conventional apatite concentration process to the residues of a concentrator, it was not attempted to optimize the flotation conditions. However, the conditions that lead to the results of Table 7 would obviously be an excellent center point for a DOE aiming at increasing the apatite recovery and reducing the consumption of the reagents, particularly the acid used to maintain the pH at 5.0 in the presence of the carbonate minerals.

The $CaO/P_2O_5$ ratio of the inverse flotation concentrate is 1.2, which is below the 1.6 target for a commercial concentrate (see Table 2). The $Al_2O_3$ and $Fe_2O_3$ constraints are also verified for the inverse flotation concentrate. However, the MgO content of 3.3% is still above the 1% target. Cleaning flotation tests were conducted on the inverse flotation rougher concentrate [18] in an attempt to bring the MgO content on the target. However, no satisfactorily results were obtained using flotation [18]. It was then necessary to develop a leaching process to upgrade the apatite concentrate by selectively dissolving the dolomite and the other carbonate minerals.

### 3.3. Leaching of the Apatite Concentrate

This section describes the tests conducted to find the leaching conditions that are favorable to the selective dissolution of dolomite $(Ca,Mg)(CO_3)_2$ in order to transfer magnesium as $Mg^{2+}$ into an aqueous phase that can be separated from the unreacted solid apatite. The acid dissolution of dolomite is given by

$$(Mg, Ca)(CO_3)_2 + 4H^+ \rightarrow Mg^{2+}_{aq} + Ca^{2+}_{aq} + 2CO_2(g) + 2H_2O \tag{1}$$

The considered leaching agents are sulfuric, nitric, hydrochloric, and acetic acids. Details of the leaching tests with the four (4) acids are given in [18]. The leaching is conducted in a mechanically agitated beaker. Acid is continuously added to the slurry of the apatite concentrate produced by inverse flotation to maintain a target pH for which dolomite is selectively attacked. Several pH levels were scanned [18] to find that a pH of 3.0 allows a maximum dissolution of dolomite with a minimum dissolution of apatite. Once the leaching is completed, the slurry is filtered to separate the solids (mainly apatite) from the aqueous phase that contains magnesium and calcium. As the acid digestion of the carbonates is exothermic the temperature remains at ~35 °C during the 60 min of reaction. These conditions reduce the formation of gypsum $(CaSO_4)$ when sulfuric acid is used for leaching. The problem of gypsum formation being overcome by the selected leaching conditions, the use of sulfuric acid is recommended as it is cheaper than HCl or $HNO_3$ that gave comparable results [18]. Although recommended by some researchers [25], the acetic acid gave a poor dissolution of the carbonates [18]. Table 8 gives the results of a $H_2SO_4$ leaching test conducted at the selected pH condition. The phosphate content of the leached product is above the target 30% $P_2O_5$ and the MgO

content is below the 1% constraint (Table 2). The mineral compositions of the flotation and leached concentrate as calculated using Whiten's approach [16] are given in Table 9. Less than 3% of the apatite is dissolved during the leaching process [18]. Results seem to indicate that dolomite and ankerite are the carbonates mainly removed by the leaching step. A phosphate concentrate with a $P_2O_5$ content above 32%, as obtained by the leaching step, can generate a selling bonus for the apatite concentrate [9,14,15], which is favorable to the economics of the process. The obtained phosphate concentrate complies with all the constraints for a commercial phosphate concentrate. As radio nuclei often follow phosphate minerals, the solid residues from the leaching has been subjected to a radio nuclei search [26] and none of the decayed elements from uranium or thorium disintegration, particularly Ra 226 and Ra 228 were detected above 0.05 mBq, in the produced phosphate concentrate. The value of the phosphate concentrate at the time of undertaking the project was estimated at 152 US$/ton [16] including a bonus [9,18] for the phosphorous content above 32% $P_2O_5$ added to the 120 US$/ton market value of 2016 (see Figure 5).

**Table 8.** Results of leaching tests with 60 min leaching at pH 3.0 ratio S/L 1/9 (average ± standard deviation of the results of 3 leaching tests).

| Species | Reverse Flotation Concentrate | Final Phosphate Concentrate | Target for a Commercial Concentrate |
|---|---|---|---|
| Weight % | 100 | 74.4 | |
| %MgO | 3.63 ± 0.6 | 0.45 ± 0.05 | <1% |
| %Al$_2$O$_3$ | 0.19 ± 0.05 | 0.21 ± 0.03 | <2% |
| %P$_2$O$_5$ | 27.32 ± 2.3 | 36.4 ± 2.9 | >30% |
| %SiO$_2$ | 0.35 ± 0.05 | 0.4 ± 0.05 | <2% |
| %CaO | 32.4 ± 3.1 | 31.7 ± 2.9 | |
| %Fe$_2$O$_3$ | 2.12 ± 0.9 | 0.96 ± 0.10 | <2% |
| %CaO/%P$_2$O$_5$ | 1.2 ± 0.4 | 0.9 ± 0.2 | <1.6 |

**Table 9.** Mineral contents of the final phosphate concentrate (average ± standard deviation of the results of 3 runs of the whole process).

| | Content (%) | |
|---|---|---|
| Species | Reverse Flotation Concentrate | Final Phosphate Concentrate |
| Apatite | 65 ± 4 | 86 ± 3.7 |
| Dolomite | 16 ± 3 | 2 ± 0.9 |
| Ankerite | 9 ± 2 | 5 ± 1.5 |
| Calcite | <1 | <1 |

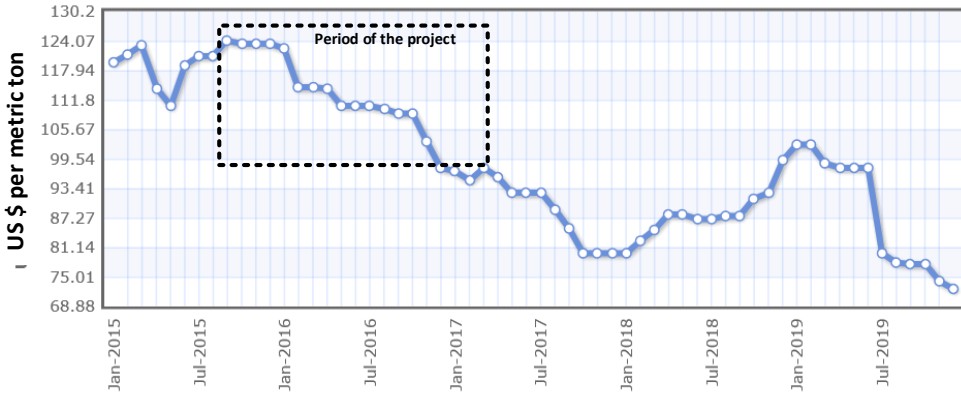

**Figure 5.** Price of phosphate concentrate [27].

### 3.4. Overall Process for the Production of a Phosphate Concentrate

The proposed desliming, inverse flotation, and leaching process yields a phosphate concentrate assaying more than 32% $P_2O_5$ with a phosphate recovery of 42%. These results compare well to those reported for a similar study [7] that yielded a phosphate concentrate assaying 29.4% $P_2O_5$ at a phosphate recovery of 46.2%.

The apatite concentration circuit is shown in Figure 6 together with the existing Niobec circuit. Before the apatite circuit is to be put into operation, it would be necessary to test for any potential harmful effect that the water additional content of $Ca^{2+}$, $Mg^{2+}$, and $PO_4^{3-}$ due to the leaching of the apatite flotation concentrate may have on the upstream carbonates and/or pyrochlore flotation circuits, as ~75% of the water is recycled from the tailings ponds back to the mill. However, a preliminary mass balance calculation indicates that the leaching of the dolomite from the apatite concentrate would cause an increase of less than 5 mg/L of $Ca^{2+}$ and $Mg^{2+}$ in the mill water, which may already contain 300 mg/L of $Ca^{2+}$ so the impact should be negligible. It was not possible to track the change of $PO_4^{3-}$ concentration in the mill water using the data available from the leaching tests. A possible water treatment plant could be considered to process the combined acid water from the pyrochlore dewatering and the water from the apatite dewatering as it is shown in Figure 6 with the objective of reducing the $Ca^{2+}$, $Mg^{2+}$, and $PO_4^{3-}$ contents before discharging the stream to the tailings ponds. This option will only be investigated if the levels of these ionic species become problematic for the recirculation of the tailings water back to the mill.

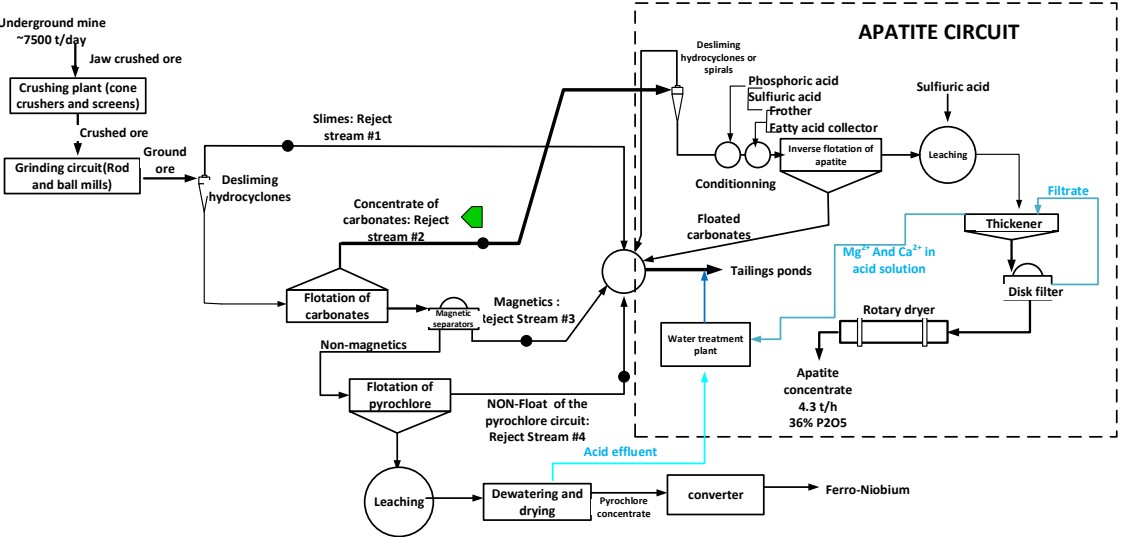

**Figure 6.** Niobec circuit including the proposed circuit for apatite concentration.

A summary of the operating conditions suggested to produce the phosphate concentrate from the concentrate of carbonates is given in Table 10. The estimated operating costs [18] are given in Table 11. The main cost item is due to the reagents, particularly the sulfuric acid consumed to maintain the pH of 4.0–5.0 during the inverse flotation of apatite and for the leaching of dolomite from the inverse flotation concentrate. The total operating cost is estimated at 143 US$/ton of concentrate. Details of the estimation of the capital costs for the project are given in [18]. The selection, design, and capital costs for the dewatering equipment were based on the actual dewatering circuit used for the pyrochlore concentrate (see Figure 1) that is also coming from an acid leaching stage with a comparable throughput to the anticipated phosphate concentrate. The operating cost is below the estimated 2017 value of 150 US$ for one ton of concentrate with the characteristics given in Table 8. Although the considered process is not yet optimized, these preliminary results show that the process could be economically viable depending of the market value for phosphate concentrate.

**Table 10.** Summary of the operating conditions for the processing of the CC.

| Processing Step | Condition | Alternative | Remark |
|---|---|---|---|
| Desliming | Hydrocyclone 5 cm diameter 0.5 cm apex Feed% solids: 15% Pressure: 140 kPa. | Spiral WWWPlus 70 cm diameter 20 L/min wash water Feed% solids: 15% | Lot of room for optimizing the process |
| Flotation | Inverse pH: 4.5 Depressant: 1.5 kg/t of $H_3PO_4$ pH modifier: $H_2SO_4$ Collector: 450 g/t of tall oil (Sylfat FA-2) Frother; 15 g/t of MIBC Flotation time (7 min) | | Important consumption of $H_2SO_4$ due to the carbonates |
| Leaching | Acid: $H_2SO_4$ Maintain a pH of 3.0 during 30 min | | |

**Table 11.** Estimated operating costs [18] for the apatite concentration circuit from the CC (reagent and energy prices are those of 2017).

| Item | $ US/ton Concentrate | % |
|---|---|---|
| Pre-concentration | 1.83 $ | 1.3 |
| Reagents | 103.86 $ | 72.3 |
| Energy | 8.38 $ | 5.8 |
| Working capital | 2.65 $ | 1.8 |
| Salaries | 21.97 $ | 15.3 |
| Concentrate dewatering & shipping | 4.95 $ | 3.4 |
| Total | 143.63 $ | 100 |
| Revenues per ton | 151.20 $ | |
| UnItary profit | 7.57 $ | |

At a price of 145 US$/ton of phosphate concentrate, the proposed apatite circuit can generate the revenues to cover the associated operating costs and allow the niobium plant to reduce its environmental footprint by taking out some of the apatite from the tailings. However, there is no plan to commission the apatite circuit in the short term as the primary focus of the Niobec personnel is currently put on the optimization of the niobium production.

## 4. Conclusions

This paper shows that a salable phosphate concentrate can be produced by processing the tailings of a niobium concentrator by applying the conventional desliming/flotation scheme used for phosphate ores. It is the first time that such demonstration is presented in the literature. For the particular case of processing a reject made of a flotation carbonate concentrate, it is recommended to use the reverse flotation process, with phosphoric acid to depress the apatite and tall oil as a collector for the carbonate minerals, to produce a phosphate concentrate assaying more than 27% $P_2O_5$. Finally, the concentrate is leached for 30 min using sulfuric acid at a pH of 3.0 to dissolve dolomite and bring the MgO content below the required constraint of 1% for a commercial phosphate concentrate. The $P_2O_5$ content of the leached concentrate is above 32%. The apatite recovery is low at 41%, but could be improved by optimizing the desliming stage. The circuit operating costs of 142 US$/ton of concentrate could be reduced by revisiting the acid requirements for the inverse flotation and final leaching. The commissioning of the proposed circuit would reduce the environmental footprint of the actual plant operation and could generate revenues. While phosphate concentrate is usually the

only product of an apatite mining operation, this paper shows that the conventional operations of desliming and flotation complemented by a hydrometallurgy process can be applied to produce a salable phosphate concentrate from the tailings of a concentrator.

**Author Contributions:** A.C. planned and carried out all the experimentation and reviewed the paper. C.B. wrote the paper and managed the funding of the project. D.D. and J.-S.M. reviewed the paper and provided the technical support on the site of mine Niobec. All authors have read and agreed to the published version of the manuscript.

**Funding:** The authors thank Niobec Mines for the technical and financial support for the realization of the project. The authors are also grateful to the FRQNT for the financial support through research grant2017-MI-202881.

**Acknowledgments:** The authors acknowledge the technical support of Vicky Dodier during the experimental test work.

**Conflicts of Interest:** The authors declare no conflict of interest.

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
