# Peer review of "Production of a Phosphate Concentrate from the Tailings of a Niobium Ore Concentrator"

_minerals, doi:10.3390/min10080692_

Round 1
Reviewer 1 Report
It took us some time to reach the aim. I am glad, I can write the advice to the Editor "accept the manuscript".
Author Response
No modification is required.
Reviewer 2 Report
In the present revised manuscript, authors have incorporated few of the comments raised during last review. However, there are several comments needs to be incorporate. Author is doing only incremental change and request to incorporate all the raised comments. The article needs revision prior to publication, and few of the comments has listed for the reference.
- Introduction: Last paragraph: Expand this paragraph and mention the methodology adopted in the present research. Please ensure to include the innovative step adopted in the present research.
- Why many places, authors have underlined the text? Please remove. For example: Table 6, 10, etc.
- Better connect your research findings to previous works published in apatite processing and characterisation.
- Still there are many grammatical errors. Request authors to review the whole manuscript with a native speaker.
- Table 6: Author should not mention the flotation mass recovery in weigh. Better convert into Wt.%.
- Figure 4 and 7: The text mentioned in the flowsheet are not readable. Please revise the font style and size for better readability.
- Figure 3: Why authors feel that this figure is important. Better delete..
- Materials and Methods: Why authors have selected only spiral concentrator and flotation for beneficiation? Why not enhanced gravity separation? Also, cite with appropriate references in that section.
- Materials and Methods: Section 3.1: Authors have mention the application of spiral is to de-slime. For the knowledge of authors, spiral should not be used as a de-sliming unit nor is meant for that. You have different classifiers including cyclones for the same application with enhanced efficiency. Please look into this seriously and revise accordingly.
- Regarding beneficiation experimentation (spiral and flotation), nothing has been explained in this section. The author must spell out the criteria on selection of variables and optimum level.
- Results and Discussion: Author must include characterisation of beneficiated and leaching products.
- References: There are many articles on this subject. The author needs to review all of these published articles on the relevant subject. Few of the relevant articles are available in the same journals also and request to include in the manuscript appropriately.
Author Response
Italic: reviewer’s comments
Bold: Author’s Reply
Small characters : Modifications or addition to the text
Introduction: Last paragraph: Expand this paragraph and mention the methodology adopted in the present research. Please ensure to include the innovative step adopted in the present research.
The following discussion was added to the text:
Many papers deal with the separation of apatite by flotation from carbonate ores[2-7] . All these papers examined the apatite concentration process from a freshly ground ore to obtain a phosphate concentrate as a primary product. Only one paper was found to deal with the production of a phosphate concentrate from the tailings of a Brazilian fertilizer plant[8]but it was not possible to find any paper describing the recovery of apatite from the tailings of a mineral processing plant which is the subject of this paper. The processing scheme usually retained for phosphate ore is grinding, desliming [2,4] to remove fine particles followed by a direct [3] or reverse [2] flotation of apatite. A good review of the flotation practices and reagents used for phosphates ores is given in [10]. This paper shows that it is possible to apply the conventional apatite ore processing scheme to produce a salable phosphate concentrate from a tailing stream of a niobium ore concentrator that recovers pyrochlore and rejects apatite with the other gangue minerals to tailings.
- Why many places, authors have underlined the text? Please remove. For example: Table 6, 10, etc.
All the underlining has been removed.
- Better connect your research findings to previous works published in apatite processing and characterisation.
We have tried but there is very little practical information regarding apatite recovery from the tailings of a mineral processing plant. When possible we have included comparison to results from other authors.
- Still there are many grammatical errors. Request authors to review the whole manuscript with a native speaker.
This will be done as soon as there would be no more major or minor adjustments to the manuscript
- Table 6: Author should not mention the flotation mass recovery in weigh. Better convert into Wt.%
Done see below…
.
- Figure 4 and 7: The text mentioned in the flowsheet are not readable. Please revise the font style and size for better readability.
We have revised the font style of Figs. 4 and 7.
- Figure 3: Why authors feel that this figure is important. Better delete..
We disagree. The figure does not use a lot of space and could help the reader that is not familiar with the operation of spirals
- Materials and Methods: Why authors have selected only spiral concentrator and flotation for beneficiation? Why not enhanced gravity separation? Also, cite with appropriate references in that section.
In a previous reply we have already replied to this comment and modified the manuscript as below:
- Materials and Methods: Section 3.1: Authors have mention the application of spiral is to de-slime. For the knowledge of authors, spiral should not be used as a de-sliming unit nor is meant for that. You have different classifiers including cyclones for the same application with enhanced efficiency. Please look into this seriously and revise accordingly.
We are currently commissionning a set-up to test various types of hydrocyclone but unfortunately at the time of the project, the set-up was not operational, and we only have limited results for various hydrocyclone diameters. However at the time of the project the set-up for the spiral was fully operational. We still think that the use of a spiral could be an innovative way to do upgrading. Indeed the spiral provides more adjustable variables (cutter opening, splitter position…) to control the quality of the separation than the hydrocyclone does.
- Regarding beneficiation experimentation (spiral and flotation), nothing has been explained in this section. The author must spell out the criteria on selection of variables and optimum level.
The text was modified as follow…
- Results and Discussion: Author must include characterisation of beneficiated and leaching products.
It was done in Tables 8 and 9. We were concerned only by the specifications of a commercial apatite concentrate. Unfortunately this is the only characterization data that we can provide:
- References: There are many articles on this subject. The author needs to review all of these published articles on the relevant subject. Few of the relevant articles are available in the same journals also and request to include in the manuscript appropriately.
We have added several of the references suggested by the reviewer. May be we have missed a paper but there are many papers on apatite flotation and we have used only two or three of them to help us in the selection of the flotation reagents.
This manuscript is a resubmission of an earlier submission. The following is a list of the peer review reports and author responses from that submission.
Round 1
Reviewer 1 Report
Dear Authors,
I’m glad because of:
- The Authors didn’t give up and resubmited the rejected article with major changes,
- The Editor sent me back the manuscript to be reviewing,
- The Authors followed the recommendations and made the manuscript more scientific i.e. the aim, the methods were significantly improved.
I find the topic of the manuscript as an interesting, utilitarian and underestimated in research. There is a one more advantage of the reviewed text. It is archiving the present technology, the generation coming after us will be able to see the state of art in a large scale mining technology.
However I find the Authors (the text) stopped in half way. The text is not finish. Please try to answer following questions:
- how the 40 kg CC sample was taken? Lines 93-96. What is a daily sample? How is being taken?
- What is the homogeneity of sampled material? What is the daily CC amount managed? Please explain in several sentences.
- The experiment and the aim. I understand, you are creating new self method including the experiment to “assesses the potential of applying the conventional apatite ore processing scheme to produce a salable phosphate concentrate from one of the tailing streams of the concentrator of the Niobec mine that exploits and processes a niobium ore to recover pyrochlore and rejects apatite to tailings with the other gangue minerals” (lines 38-40). Please write several sentences to explain.
- Consider adding a paragraph to the manuscript discussing how yours invention will influence the tailings waste generation process?
My recommendation is accept the manuscript after the major corrections.
Reviewer 2 Report
In the present revised manuscript, authors have incorporated most of the comments raised during last review. However, there are few areas needs minor revision. The article needs revision prior to publication, and few of the comments has listed for the reference.
- Introduction: Last paragraph: It is not necessary in a research paper; It is usual practice in report or thesis. So, request to authors to please remove this paragraph.
- Plant data should be always in arrange. Author should mention the variation as well as standard deviation in Tables (1, 2, 4).
- Better connect your research findings to previous works published in apatite processing and characterisation.
- The main drawback of this paper is the lack of proper discussion and scientific approach during experimentation. The discussion section needs to be strengthened with the available literature, characterisation and the innovative results achieved.
- The innovation and the importance of this work are not highlighted in the abstract, introduction and conclusions.
- Still there are many grammatical errors. Request authors to review the whole manuscript with a native speaker.
- Table 6 and 7: Author should not mention the flotation mass recovery in weigh. Better convert into Wt.%.
- Introduction: Author should narrate the beneficiation of such tailing streams and its impact on overall recovery.
- Materials and Methods: Why authors have selected only spiral concentrator and flotation for beneficiation? Why not enhanced gravity separation? Also, cite with appropriate references in that section.
- Materials and Methods: Figure 3: Is this figure required? Better connect with citation with your old work for reference.
- Materials and Methods: Section 3.1: Authors have mention the application of spiral is to deslime. For the knowledge of authors, spiral should not be used as a desliming unit nor it is meant for that. You have different classifiers including cyclones for the same application with enhanced efficiency. Please look into this seriously and revise accordingly.
- Regarding beneficiation experimentation (spiral and flotation), nothing has been explained in this section. The author must spell out the criteria on selection of variables and optimum level.
- Materials and Methods: Author must indicate the error involved in the measurement system and obtained readings.
- Results and Discussion: Author must include characterisation of beneficiated and leaching products.
- Results and Discussion: The economic aspect must cite with the source of reference and along with the date.
- References: There are many articles on this subject. The author needs to review all of these published articles on the relevant subject. Few of the relevant articles are mentioned below for your reference and request to include in the manuscript appropriately.
- Feng , D. and Aldrich , C. , 2004 , “Influence of operating parameters on the flotation of apatite.” Minerals Engineering , 17 , pp. 453 – 455
- Ni, X., Parrent, M., Cao, M., Huang, L., Bouajila, A., & Liu, Q. (2012). Developing flotation reagents for niobium oxide recovery from carbonatite Nb ores. Minerals Engineering, 36, 111-118.
- Dehghani, A., Azizi, A., Mojtahedzadeh, S. H., & Gharibi, K. (2012). Optimizing rougher flotation parameters of the Esfordi phosphate ore. Mineral Processing and Extractive Metallurgy Review, 33(4), 260-268.
- Yu, B., & Aghamirian, M. (2015). REO mineral separation from silicates and carbonate gangue minerals. Canadian Metallurgical Quarterly, 54(4), 377-387.
- Wang, L., Tian, M., Khoso, S. A., Hu, Y., Sun, W., & Gao, Z. (2019). Improved Flotation Separation of Apatite from Calcite with Benzohydroxamic Acid Collector. Mineral Processing and Extractive Metallurgy Review, 40(6), 427-436.
- Farzanegan, A., & Mirzaei, Z. S. (2015). Scenario-Based Multi-Objective Genetic Algorithm Optimization of Closed Ball-Milling Circuit of Esfordi Phosphate Plant. Mineral Processing and Extractive Metallurgy Review, 36(2), 71-82.
- Tripathy, S. K., & Murthy, Y. R. (2012). Multiobjective optimisation of spiral concentrator for separation of ultrafine chromite. International Journal of Mining and Mineral Engineering, 4(2), 151-162.
- Liu, X., Zhang, Y., Liu, T., Cai, Z., & Sun, K. (2017). Characterization and separation studies of a fine sedimentary phosphate ore slime. Minerals, 7(6), 94.
- Bazin, C., Sadeghi, M., Bourassa, M., Roy, P., Lavoie, F., Cataford, D., ... & Gosselin, C. (2014). Size recovery curves of minerals in industrial spirals for processing iron oxide ores. Minerals Engineering, 65, 115-123.
- Tripathy, S. K., Murthy, Y. R., Singh, V., Farrokhpay, S., & Filippov, L. O. (2019). Improving the Quality of Ferruginous Chromite Concentrates Via Physical Separation Methods. Minerals, 9(11), 667.
- Sadeghi, M., Bazin, C., & Renaud, M. (2014). Effect of wash water on the mineral size recovery curves in a spiral concentrator used for iron ore processing. International Journal of Mineral Processing, 129, 22-26.
- Tripathy, S. K., & Murthy, Y. R. (2012). Modeling and optimization of spiral concentrator for separation of ultrafine chromite. Powder technology, 221, 387-394.
- Liu, X., Zhang, Y., Liu, T., Cai, Z., Chen, T., & Sun, K. (2016). Beneficiation of a sedimentary phosphate ore by a combination of spiral gravity and direct-reverse flotation. Minerals, 6(2), 38.
Reviewer 3 Report
The manuscript described the approach taken to produce a phosphate concentrate as a secondary product from the tailings of a niobium ore concentrator. It was proposed that the reverse flotation of apatite yielded better results than the commonly used direct flotation of apatite, and a slurry desliming followed by reverse flotation of apatite and an acid leaching of the apatite concentrate. It was reported that the obtained phosphate concentrate assays more than 32% P2O5 at a P2O5 recovery of 41%.
Some comments are listed as follows:
line 58 Figure 1. text is too small;
the material flow line is missing for "the non-floated material from the carbonate circuit passes through magnetic separators"
same problem with Figure 7.
line 239 Equation should be numbered.
line 255 "removed"
line 268 reference format
line 270 Section 3.4 overall process
The conclusions are groundless or even misleading. For example, it is claimed, as shown on line 271-272, that “The circuit is simple, uses no circulating load and should not generate any harmful reject.” Figure 7, the proposed flowsheet, clearly showed that Mg2+ and Ca2+ solution will be pumped to tailing pond finally. It is well known that the decant from tailing pond will be pumped back to the mill fro reuse. The [Mg2+] and [Ca2+] in solution will keep on building up because the leaching process will generate more and more Mg2+ and Ca2+ and there are no measures being proposed to stabilize them. These multivalent cations in solution will definitely impact the flotation of carbonates and the flotation of pyrochlore.
In addition, because the leaching solution emerging into the water system, quote, “Indeed the aqueous phase from the leaching of the inverse flotation concentrate is rejected with the floated carbonate stream from the inverse apatite flotation to ensure that any excess of acid will be neutralized”, the pH of water will keep on increasing if no additional alkali is added, because “acid will NOT be neutralized by mixing with natural water”.
As such, the impact of the introduction of Mg2+, Ca2+ and H+ species into flotation water system has to be studied before a conclusion, quotes, “The circuit is simple, uses no circulating load and should not generate any harmful reject” can be reached. Otherwise, it might bring a disaster to the flotation of carbonates and the flotation of pyrochlore.
Line 268 Figure 6 shows “the period of the project” is from 2015-2017. Has the proposed flowsheet been adopted and applied at the mine by now, that is, three years later? If not, why?
In addition, Figure 6 shows that, from 2015 to recently, the price of phosphate concentrate kept on decreasing to a bottom level. Is it still profitable to recover apatite for now? After all, as shown by Table 9, the unitary profit was only 8.57 $/ton when the price of phosphate concentrate was higher.